# Pediatric Sarcomas: The Next Generation of Molecular Studies

**DOI:** 10.3390/cancers14102515

**Published:** 2022-05-20

**Authors:** Petros Giannikopoulos, David M. Parham

**Affiliations:** 1Innovative Genomics Institute, University of California, Berkeley, CA 94704, USA; pgiannikopoulos@berkeley.edu; 2Department of Anatomic Pathology, Children’s Hospital Los Angeles, Los Angeles, CA 90027, USA; 3Department of Pathology, University of Southern California Keck School of Medicine, Los Angeles, CA 90033, USA

**Keywords:** pediatric sarcoma, molecular genetics, technology, scRNA seq, spatial multi-omics, high-throughput functional genomics, CRISPR-Cas9, specimen annotation

## Abstract

**Simple Summary:**

There has been an incredible amount of discovery in pediatric sarcomas, but much remains to be accomplished. Clinical challenges include diagnostic heterogeneity and the poor outcome of patients with high risk, metastatic, and relapsed disease. The emergence of single cell sequencing has allowed the ability to document tumor cell heterogeneity in amazing detail, but it does not allow the ability to visualize spatial orientation. This problem has been solved by spatial multi-omics, which can be used to map tumors and visualize the distribution of critical transcripts, mutations, and proteins. However, these tools only offer observational data. High-throughput functional genomics provides a powerful way to highlight oncogenic drivers and potential therapy opportunities. Research has been hamstrung by a need for annotated specimens, particularly in post-therapy, relapsed, and metastatic disease, and initial biopsies offer only limited data opportunities. Data complexity, variability, and inconsistency present problems best approached with AI/machine learning. We stand on the threshold of a revolution in cancer cell biology that has the potential for translation into more effective and more directed therapies, particularly for previously recalcitrant diseases.

**Abstract:**

Pediatric sarcomas constitute one of the largest groups of childhood cancers, following hematopoietic, neural, and renal lesions. Partly because of their diversity, they continue to offer challenges in diagnosis and treatment. In spite of the diagnostic, nosologic, and therapeutic gains made with genetic technology, newer means for investigation are needed. This article reviews emerging technology being used to study human neoplasia and how these methods might be applicable to pediatric sarcomas. Methods reviewed include single cell RNA sequencing (scRNAseq), spatial multi-omics, high-throughput functional genomics, and clustered regularly interspersed short palindromic sequence-Cas9 (CRISPR-Cas9) technology. In spite of these advances, the field continues to be challenged by a dearth of properly annotated materials, particularly from recurrences and metastases and pre- and post-treatment samples.

## 1. Introduction

Sarcomas consist of a wide array of cancers that arise in bones and soft tissues. They comprise a relatively rare form of cancer, particularly in adults where they are overshadowed by carcinomas, melanomas, and hematopoietic tumors such as leukemias and lymphomas. In children, however, they gain some ascendancy in prevalence, making up the fourth most common group of childhood cancer, following leukemia, brain tumors, and lymphoma [1]. They form a heterogeneous group composed of a wide array of tissue diagnoses, each caused by molecular alterations that vary in specificity and type. These alterations include epigenetic changes seen in embryonic tumors such as embryonal rhabdomyosarcomas, chimeric proteins found in cancers such as Ewing sarcomas, alveolar rhabdomyosarcomas, and synovial sarcomas, chromosomal deletions such as present in rhabdoid sarcoma, amplified genes as seen in alveolar rhabdomyosarcomas, and chromothripsis and genomic complexity as exemplified by osteosarcoma. Inheritance of parental mutations leading to cancer susceptibility is relatively common [2], and secondary mutations and epigenetic changes are often seen in recurrent lesions and lead to chemotherapy resistance [3].

Although there have been many major discoveries in the field of pediatric sarcoma oncology, many unanswered questions remain, as evidenced by the relatively high number of “undifferentiated sarcomas” and “sarcomas, not otherwise specified” [4]. Although gene expression profiling solves many of these mysteries, the commercial production of gene panels lags behind the rate of new discoveries. The mutational spectrum of some pediatric sarcomas is much broader than initially realized, as with Ewing-like sarcomas [5] and infantile fibrosarcoma spectrum lesions [6,7]. Another problem lies in solving the mysteries behind the directed therapy-susceptibility of some tumors and the resistance of others [8]. Solving the delivery of mutation-specific cancer delivery agents against aberrant transcription factors remains a major challenge [3]. Finally, predicting potential therapy responders and non-responders has major implications for preventing the untoward effects of therapeutic agents in growing children and finding ways to combat non-responsive, recurrent, and metastatic tumors remain challenges [3].

While recent whole exome, whole genome, and whole transcriptome sequencing studies of pediatric sarcomas have greatly advanced our understanding of this diverse set of tumors, these analyses only provide bulk averages of DNA and RNA alterations across a tumor’s constituent cells. Not only do such aggregate DNA and RNA analyses fail to reveal the full extent of the biological diversity amongst the cancerous and non-cancerous cells in a tumor, they also fail to shed light on the role of the epigenome, whose aberrations are increasingly being shown to play a central role in pediatric tumorigenesis [9,10]. Pediatric sarcomas, similar to cells undergoing differentiation during embryogenesis, are regulated by the complex interplay between transcription factors, chromatin regulators, modified histones, regulatory elements, non-coding RNAs, and more. Therefore, gaining a comprehensive view of the biology of pediatric sarcomas requires a new generation of molecular methods capable of characterizing in detail the state of the genomes, transcriptomes, and epigenomes within constituent cells of tumors, including the hierarchical three-dimensional landscape of chromosomes, which play a central role in how specific transcriptional programs are controlled [11,12,13,14].

Furthermore, even the highest resolution, multi-dimensional analyses of pediatric sarcomas represent mere observational snapshots of the molecular states in which a given tumor’s constituent cells reside. As a consequence, the significance of a set of observations must be inferred based on prior knowledge, statistical inference, or in silico models. In order to overcome this challenge of determining the biological consequence of observed genomic alterations, tremendous progress has been made in recent years in the development of “functional genomics” methods, laboratory approaches that enable the functional characterization of specific molecular alterations.

In this review we explore some of the latest, cutting-edge laboratory tools that are advancing our knowledge of the biological underpinnings of pediatric sarcomas, and we will focus specifically on how these methods are shedding light on new potential therapeutic strategies, especially for the most challenging high-risk cases.

## 2. Clinical Challenges of Pediatric Sarcomas

Sarcoma heterogeneity poses a large challenge to pediatric oncology research. The number of cases is relatively small, so that diverse lesions must be lumped together in order to reach the statistical power required for a five-year study. For the purpose of therapy, pediatric sarcomas have to this point been divided into rhabdomyosarcomas (RMS) and “non-rhabdomyosarcomatous soft tissue sarcomas” (NRSTS), since as a group, RMS comprise roughly one-half of the soft tissue cancers seen in children [15]. RMS have been similarly divided into *PAX-FOXO1* “fusion-positive” and “fusion-negative” tumors, because of the more aggressive nature of the former lesions. However, it is now becoming apparent that other mechanisms may be in play in therapy-resistant lesions, as with the *MYOD1* mutations seen with aggressive fusion-negative spindle cell RMS [15].

For treatment protocols, RMS has been divided into “low-risk”, “intermediate-risk”, and “high risk”, based on prognostic indicators such as stage, age, histology (in former protocols), and fusion status (in current protocols) [15]. A combination of multi-agent chemotherapy, comprising a backbone of vincristine, actinomycin D, and cyclophosphamide (VAC) has been utilized since the initial trials of the Intergroup Rhabdomyosarcoma Study Group (IRSG), and more effective agents have been elusive [15]. Chemotherapy has been combined with external beam radiation and newer radiotherapeutic agents, combined with post-adjuvant surgery, for local control. These have yielded 89%, 63%, and 38% three to four-year event-free survival rates for low-, intermediate-, and high-risk RMS, respectively, in the most recently published Children’s Oncology Group (COG) trials [16,17,18]. Work continues to find effective therapies for relapsed RMS, although a classification devised by Oberlin et al. [17] has been useful in predicting response in these lesions.

NRSTS has been a relative newcomer to multi-institutional trials of pediatric NRSTS, with the initial trials reported by the Pediatric Oncology Group (POG) in 1999 [19]. RMS-type therapy was used in the initial POG trial, but without any apparent effect. Grading appeared to be important in predicting outcome, and a pediatric sarcoma-specific grading scheme was devised [20]. The COG assumed this effort with ARST0332, using a combination of ifosfamide, doxorubicin, and radiotherapy. This trial used a system devised by Spunt et al. to assign patients to three risk groups, which proved to be useful in defining treatment failure risk [4]. However, heterogeneity in terms of age, primary tumor site, and histologic diagnosis posed a major limitation that may have masked therapy considerations for patient subsets. During this trial, 5-year event-free survival rates ranged from 89% for low-risk patients to 65% for intermediate risk patients to 21% for high-risk patients.

Mutation-specific therapy has been successful in eradication of some sarcomas, beginning with the discovery of the use of imatinib as targeted therapy for gastrointestinal stromal tumors (GIST). However, since many pediatric GIST are negative for c-kit/PDGFRB mutations, more effective therapy is still needed for children [21]. Crizotinib has been effective in eliminating many inflammatory myofibroblastic tumors (IMT), but responses have been unpredictable [22]. More recently, larotrectinib has been used as directed therapy against the infantile fibrosarcomas and related tumors with TRK mutations [23]. The mutational spectrum of this group grows increasingly complex, so that additional agents may be required for future directed therapy.

Bone sarcomas are slightly less common in pediatric age patients, but they nevertheless comprise a major cancer group in children. The most common bone sarcoma, osteosarcoma, has a bimodal age distribution, peaking in adolescence and older adulthood. Osteosarcomas require complex multi-disciplinary therapy, including multi-agent induction chemotherapy, surgical resection, adjuvant chemotherapy, and local control for metastases [24]. Radiation is used as an adjuvant therapy for microscopic disease or incomplete gross resection. A combination of methotrexate, Adriamycin, and cisplatin (MAP) has been the most commonly used therapy backbone. Overall survival has been around 70% after 5–6 years [24]. Because treatment intensification has failed to improve survival rates, the use of targeted therapy for recalcitrant tumors is being investigated [25]. However, osteosarcomas are biologically complex childhood tumors, characterized by genomic instability, unchecked cell cycle progression, and a propensity to develop drug resistance [25]. Osteosarcoma pathogenesis involves the Notch, Wnt, NF-κB, p53, PI3K/Akt, and MAPK pathways, as well as regulatory miRNAs [26], and chromosomal instability is characterized by chromothripsis and kataegis. Similarly, osteosarcomas possess complicated cellular microenvironments, with bone, stroma, vascular cells, and immune cells that support their growth and dissemination. Analysis of this complex microenvironment offers insight into future therapy approaches [27].

The second most common bone sarcoma, Ewing’s sarcoma, occurs most commonly in adolescents and young adults but may occur at all ages. Extraosseous Ewing sarcomas occur relatively frequently, but extraskeletal osteosarcoma is extremely uncommon in children. Although extraskeletal Ewing sarcomas were once treated with rhabdomyosarcoma therapy [28], extraosseous and skeletal Ewing sarcomas have been treated with the same COG protocol [29]. An extraskeletal location appears to be a favorable prognostic sign [30], and patients appear to benefit by being on an osseous Ewing sarcoma protocol instead of soft tissue sarcoma therapy [31]. Use of ifosfamide and etoposide as chemotherapeutic agents has led to improved survival with localized Ewing sarcoma, with response rates of around 60 to 70% [3]. However, the outlook for metastatic and relapsed disease remains poor, although high-dose ifosfamide appears to be an option for relapse therapy [3]. A number of factors affect outcome, including age, site, and histological response to therapy [3]. Targeted therapies are being investigated for relapsed disease [3]. As with osteosarcoma, tumor cell microenvironment appears to play a role in Ewing sarcoma growth and therapy resistance [3]. Ewing sarcomas contain a characteristic gene fusion between *EWSR1* and a variety of ETS family genes, most commonly *FLI1*, whose protein products offer attractive but currently impregnable targets. Similar round cell bone and soft tissue sarcomas contain either non-ETS fusions or other rearrangements, notably *CIC* or *BCOR* fusions [3].

## 3. Comprehensive Molecular Profiling at Single-Cell Resolution

Several groundbreaking technologies have been developed in recent years that enable the comprehensive profiling of the genomes, transcriptomes, epigenomes, and proteomes of single human cells [32,33,34,35,36,37,38,39], opening the door to new potential breakthroughs in our understanding of pediatric sarcoma biology. Of these technologies, single-cell RNA sequencing (scRNA-seq) has led the way in characterizing tumor cell heterogeneity, especially for pediatric cancers since the mutational burden is quite low. The advantage of scRNA-seq is that the gene expression profile of a single tumor cell most informatively reflects that cell’s functional state, which can vary markedly due to mutations, epigenetic modifications, cell cycle position, stress, lineage, cell–cell interactions, and other causes. scRNA-seq also enables the classification and functional characterization of a tumor’s non-cancerous cells, which is not possible with single-cell DNA sequencing, and cannot functionally classify and characterize non-cancerous cells because of the identical nature of their genomes [40,41,42].

While the specifics of different scRNA-seq protocols vary, they share a common strategy that consists of (1) disaggregating tumors into a suspension of single cells; (2) isolating the disaggregated cells using microfluidics, flow cytometry, or micro-droplet separation; (3) converting the RNA of each isolated single cell into cDNA using reverse transcriptase; (4) ligating a cell-specific oligonucleotide barcode to the cDNA fragments from each cell; and (5) sequencing the entire pool of barcoded cDNA via next-generation sequencing (Figure 1). By using barcode sequences, the expression data can then be computationally parsed, enabling the measurement of the complete transcriptional output of every cell [43,44].

In addition to scRNA-seq, a number of groundbreaking approaches have emerged in recent years that make it possible to study the epigenomes of pediatric sarcomas at unprecedented single-cell resolution.

Several of these next-generation epigenomic profiling methods, namely ATAC-seq, ChIP-seq, and AQuA-HiChIP were deployed by Khan and colleagues in a recent study that delineated the core regulatory circuitry of pediatric rhabdomyosarcoma (RMS) in breathtaking detail [45]. These methods enabled the investigators to show that the product of the *PAX3-FOXO1* fusion gene binds to super enhancer histone regulatory complexes associated with SOX8, and that SOX8 negatively regulates the pro-myogenic core regulatory transcription factors MYOD1 and MYOG. These discoveries indicate a histone deacetylase (HDAC)-mediated mechanistic link between the *PAX3-FOXO1* fusion protein and a halt in skeletal muscle differentiation, an insight that would have been impossible to derive through bulk DNA or RNA sequencing alone.

The impact that such single-cell datasets are beginning to have on our biological understanding of pediatric tumors cannot be overstated. A recent large collaborative study examined the transcriptomes of nearly 9000 single cells from 25 medulloblastomas spanning all four molecular subtypes (SHH, WNT, Group 3, and Group 4) identified multiple distinct malignant cell populations within each subtype, providing the first ever cellular atlas of medulloblastoma [46]. The study also delineated the developmental origins leading to the biological differences between each molecular subtype.

Similar approaches are now beginning to be applied in other pediatric tumors, including sarcomas [45,47,48,49,50,51,52,53]. A study of synovial sarcoma using an integrative approach with scRNA-seq and spatial profiling revealed a malignant subpopulation marking immune-deprived regions and predicting poor clinical outcome. Further study revealed that this population could be repressed by immune cytokines and targeted with HDAC and CDK4/CDK6 inhibitors that enhanced tumor cell immunogenicity [54]. In another study, Miller et al. generated single cell transcriptomes of Ewing sarcoma cell lines and merged them with other published profiles to uncover subpopulations enriched for mesenchymal markers and low EWSRI-FLI1 expression [55]. Similarly, a recent scRNA-seq study of advanced osteosarcoma revealed distinct patterns of intratumoral heterogeneity between primary, metastatic, and recurrent osteosarcoma specimens [56]. A recent report by Patel et al. [57] used scRNAseq to study how chemotherapy affects myogenesis and post-therapy tumor cell reconstitution in embryonal rhabdomyosarcoma. This study suggested that targeting the developmental stage offers an effective stratagem in preventing recurrence of that tumor. These findings all underscore the importance of tumor cell heterogeneity to newer therapy approaches.

## 4. Spatial Multi-Omics

One of the limitations of single-cell sequencing is that it requires tumor cells to be disassociated and then lysed prior to analysis. Current single-cell sequencing methods are therefore inherently incapable of providing any information regarding the intratumoral location of individual cells or the intracellular location of specific analytes such as transcripts or protein. Given the important biological roles that spatial gene and protein expression heterogeneity plays in various physiologic and pathologic contexts [58,59], significant efforts have been made by multiple groups in recent years to develop high-resolution, comprehensive spatial maps of proteins, transcripts, and the genome [60,61,62,63,64,65,66,67,68,69,70,71,72,73,74,75,76].

Two broad categories of methods provide parallel in situ characterization of the intracellular and intercellular spatial orientation of large numbers of biomarkers via high-resolution fluorescence microscopy, though their application to pediatric sarcomas has been limited to date. The core strategy of one category partitions tissue sections into an ultrafine two-dimensional grid and, by using techniques such as mass spectroscopy, immunofluorescence, or next-generation sequencing, performs a massively multiplexed assay on the tissue fragment at each “pixel” of the grid. After storage of assay results for each pixel, a spatial biomarker map computationally reconstructs the biomarker milieu of the tissue (Figure 2). Two prominent recent examples of this strategy are “Slide-seq”, a method that can quantify the spatial distribution of RNA transcripts comprising the whole transcriptome in single tissue section [72], and “Slide-DNA-seq”, a similar approach that enables spatially resolved DNA sequencing within intact tissue sections [77]. The second broad strategy for spatial multi-omic profiling relies on differential labeling of large numbers of probes such as antibodies or in situ hybridization probes followed by the microscopic visualization of those labels. Using fluorescence tagging for differential labeling, sections incubated with fluorescently labeled probes indicate the location and abundance of each analyte via high-resolution fluorescence microscopy (Figure 2). These two general strategies (high-resolution, cyclic fluorescence imaging versus multiplexed molecular analysis of partitioned tissue section) are not mutually exclusive, as evidenced by the recent emergence of “in situ genome sequencing” or “ISH”, whereby thousands of genomic loci within a single nucleus can be spatially localized and fully sequenced [78].

In contrast to RNA, for which it is now possible to spatially map the entire transcriptome within individual cells [64,72], the spatial profiling of proteins has proven to be somewhat more challenging, largely due to the inherent limitations of antibodies and mass spectrometry [79], and due to the fact that the abundance of individual proteins within single cells can span over seven orders of magnitude [80]. However, advancements in protein chemistry, imaging, and data processing are rapidly increasing the number of proteins that can be spatially characterized [81]. New tools are emerging at an accelerating pace, and collectively, offer unprecedented promise for pediatric sarcoma research. For example, a recent study of chordoma, a mesenchymal tumor of bone with aggressive behavior, utilized a combination of RNA-seq and imaging. In this comparison of chordoma and benign nucleus pulposus, microenvironmental differences in expression of certain proteins, particularly carbonic anhydrase II, appeared to facilitate cell growth and migration and could be targeted for cancer inhibition [82].

Characterization of spatial relationships of protein expression in pediatric sarcomas offers significant advantages in the study of pediatric sarcomas, in order to explore connection between cell and tissue development and disease progression and how genetic changes affect cellular relationships [83]. Heterogeneity of the tumor cell microenvironment constitutes a major factor in a variability of these spatially complex neoplasms, such as osteosarcoma where tumor cells interact with stromal cells to support growth and metastasis [56]. Even in less obviously complex tumors such as Ewing sarcoma, stromal interactions give rise to differences in gene expression signatures, methylation profiles, and fusion gene expression [3]. The ability to visual gene expression may assist us in understanding the relationships between treatment response and the cellular microenvironment. However, these techniques are limited by the relatively small number of proteins that can be visualized on a single cell, as well as the problems inherent with small biopsies and limited samples that may neglect important components in a large tumor.

## 5. High-Throughput Functional Genomics

As illuminating as the aforementioned genomic profiling technologies have proven in recent years, they can only provide a static view of the biological state of the cells within a sarcoma patient’s tumor, often making the interpretation of the alterations observed quite challenging. To overcome this challenge, investigators frequently simulate an observed set of alterations in a model system (i.e., cell line or animal model), referred to as the “perturbation,” and then monitor its biological impact, referred to as the “phenotype.” This kind of targeted, hypothesis-driven, perturbation-to-phenotype empirical analysis has been the workhorse of cancer research for decades [84,85].

However, validating molecular alterations one perturbation at a time is a tedious process that requires inference by use of prior knowledge and observations. In response to these bottlenecks, high-throughput genetic screening was developed. The basic strategy of high-throughput screening consists of performing thousands of parallel cellular perturbations, and then using defined phenotypic criteria (e.g., cell death, cell growth, specific gene expression changes, etc.) to identify biologically significant alterations. In essence, high-throughput screening is the reverse of the classical approach, where the perturbation is known and the phenotype is unknown. In high-throughput screening, the phenotype (i.e., the “screen”) is known, while the perturbation (i.e., the causal mechanism of the desired phenotype) is unknown.

The methods used to perturb cells in the earliest generation of high-throughput screens included DNA mutagenesis, exposure to pharmacodynamically active drugs, and overexpression of genes via transfection or viral transduction [86,87,88]. While immensely impactful, these initial screening strategies exhibit a number of shortcomings, the most significant of which is lack of specificity. A mutagen, drug, or virus may introduce a diverse array of different perturbations within a single cell, making delineation of the true cause of the phenotype a difficult task [89].

High-throughput genetic screening enjoyed a massive improvement following the advent of genome-scale RNA interference (RNAi), and even more so following the subsequent discovery of the bacterial adaptive immune system CRISPR (clustered regularly interspersed short palindromic sequences) and its associated RNA-guided endonuclease Cas9 [90,91,92,93,94,95]. By allowing precise perturbation of any RNA sequence in the genome, these tools officially herald the era of large-scale “functional genomics”.

Re-engineering of the CRISPR/Cas9 system enables the induction or repression of any gene’s expression, and high-throughput genetic screening using this tool offers a promising advancement for pediatric sarcomas. In 2013, investigators at the University of California, San Francisco (UCSF), and the University of California, Berkeley, demonstrated use of the CRISPR/Cas9 system for ultraspecific control of gene expression [96], using an approach referred to as CRISPR-inhibition (“CRISPR-i”) and CRISPR-activation (“CRISPR-a”). In a seminal study published in 2014, the same group published the results of a groundbreaking genome-wide CRISPR-i/a screen, which assessed the cellular consequences of systematically modulating the levels of specific transcripts [97]. In 2021, this innovation was followed by a landmark optimization of the CRISPR-i/a approach, “CRISPRoff,” whereby the Cas9 protein is modified to enable the programmable epigenetic silencing of any part of the genome [98]. Given the role that dysregulated transcription plays in pathogenesis of pediatric sarcomas [99,100], these powerful new tools for high-throughput genetic screening have the potential to transform the therapeutic landscape for pediatric sarcomas.

Though the initial CRISPR-i/a screens relied on bulk phenotypic characterizations such as cell growth and cell death, the field took a major step forward when single-cell RNA-seq was used as the phenotypic readout of single and multiple (i.e., combinatorial) perturbations [101,102,103]. This method, referred to as “Perturb-seq” now enables scientists to obtain a comprehensive biological snapshot, at the single-cell level, of thousands of specific gene expression perturbations in one experiment (Figure 3). These cutting-edge functional genomics screening tools are now in the initial phases of being deployed in pediatric malignancies. For example, a genome-scale CRISPR-Cas9 screen in Ewing sarcoma revealed multiple therapeutic targets in wild-type *TP53* cases [104]. More recently, a large-scale CRISPR-based, perturbational screen in a panel of malignant rhabdoid tumor (MRT) cell lines showed that the downstream signaling pathways of different receptor tyrosine kinases, namely PDGFRA and MET, are activated in MRT and represent therapeutic targets [105]. The proliferation of such studies is only accelerating [106,107,108], and represents a significant source of future optimism.

In a particularly exciting recent development, CRISPR-based functional in vivo genomic screens have been specifically applied to mouse models [109,110,111,112,113,114,115,116,117,118,119]. The general approach consists of: (1) selecting a relevant mouse model for a specific cancer type, typically a genetically engineered mouse or a patient-derived xenograft system (PDX); (2) performing high-throughput mutagenesis via coordinated delivery of Cas9 protein and guide RNAs (sgRNAs); (3) selecting cells exhibiting the desired phenotype; and (4) delineating the underlying genetic alteration induced by the screen. The advantage of using an in vivo screen versus an in vitro cell-line-based screen is the higher degree of biological relevance that in vivo models afford [120].

## 6. Need for Clinically and Pathologically Annotated Specimens

As new and increasingly powerful laboratory technologies emerge, a major factor that may limit the clinical impact that they can make is access to specimens that are most relevant in this clinical context, i.e., tissue from relapsed and metastatic lesions. The vast majority of the sarcoma samples analyzed in recent comprehensive genomic studies have been primary, pre-treatment specimens. For example, the COG Ewing sarcoma tissue banking protocols AEWS02B1 and AEWS07B1 accrued tissue in only about 18% of patients, and among these specimens only 29% comprised fresh, frozen tissues and only 7 were metastatic [121]. Another problem has been a lack of clinical annotation from patients not enrolled in a concurrent therapeutic trial, as attested by this occurrence in 95% of samples obtained in the COG osteosarcoma biology study P9951. This discrepancy was resolved after forming an office providing infrastructure and a linkage with annotated patient data [122]. A further problem has been the variability of the samples received, as shown by a COG of predictors of survival in Ewing sarcoma. Gene-specific enrichment analysis revealed that samples with stromal contamination demonstrated a gene signature associated with survival, whereas samples without stromal contamination did not [123].

The few studies that have comprehensively analyzed paired pre- and post-treatment tumor specimens suggest that, similar to adult cancers, primary pediatric malignancies comprise molecularly distinct subpopulations of cells that undergo clonal divergence following exposure to the selective pressure of cytotoxic therapy. For example, a recent whole-genome sequencing analysis of 33 paired diagnostic and post-treatment medulloblastoma specimens revealed that recurrent tumors retained only 12% of the somatic alterations identified in their primaries [124]. This finding highlights the importance of assessing both primary and post-treatment specimens. Given the diversity of somatic alterations in pediatric sarcomas, these types of paired analyses will be critical for the development of more effective treatments for high-risk patients.

## 7. Artificial Intelligence/Machine Learning

In addition to the growing volume of data that the aforementioned methods will generate in the years to come, a major catalyst that stands to greatly accelerate the development of therapeutically actionable insights for pediatric sarcomas is artificial intelligence (AI) and machine learning (ML), specifically in the form of convolutional neural networks, also referred to as “deep learning” [125,126]. By recognizing and classifying patterns embedded in massive datasets of various types, from radiologic imaging to gene expression data [127] to somatic mutational data [128] to biophysical spectra [129] to routine histolopathologic images [130,131,132], deep learning can produce powerful predictions that humans are simply incapable of generating due to the inherent limitations of our biology [133]. Furthermore, with the recent landmark release of AlphaFold, a highly accurate, deep-learning-based, protein-structure prediction tool [134], and the proliferation of AI-based drug discovery tools [135], AI is positioned to not only rapidly increase our understanding of pediatric sarcomas, but also give rise to novel effective treatment strategies that otherwise would have remained undiscovered.

One of the challenges of applying AI to pediatric sarcomas is their relative rarity and heterogeneity, both intratumoral and intertumoral (i.e., case-to-case variability). Intratumoral cell heterogeneity of pediatric sarcoma poses challenges to the effective use of machine learning, because the initial biopsy only contains a limited sample of tissue and may underestimate factors such as tumor grade or treatment effect. For this reason, machine learning for sarcoma research to date has been used most effectively in radiology. For example, one group recently reported that MR images of soft tissue sarcomas of varying grades could be used to create a validated data set in which a combination of features yielded accuracy and specificity of >90% [136]. Another study used machine-learning to delineate tissue components of soft tissue sarcomas in order to monitor post-therapy changes [137], and in a similar vein, other groups trained AI algorithms to assess post-therapy necrosis of osteosarcoma using MR images [138] and predict survival in high-grade soft tissue sarcomas using “radiomics” [139]. Machine learning models have also been shown to be useful in predicting metastasis from soft tissue sarcoma [140], and to outperform humans in separating malignant from benign malignant peripheral nerve sheath tumors [141]. Furthermore, in addition to predicting survival and recurrence, groups have demonstrated the feasibility of using AI to delineate gross tumor volume for radiation therapy treatment planning [142].

With respect to pathology, a growing number of groups have demonstrated that AI algorithms can not only be trained to perform as well as expert pathologists in diagnosing and grading certain common adult cancers (e.g., prostate cancer) [143,144,145], they can also predict the presence or absence of clinically significant molecular biomarkers based on morphology alone [130,146]. These tools are now being deployed in pediatric sarcomas [147,148,149,150,151]. For example, a multi-institutional effort developed a deep learning neural network-based diagnosis system to diagnose and classify rhabdomyosarcoma, suggesting that AI may be useful to assist pathologists with limited diagnostic expertise [149]. Another group recently used machine learning to develop an optimal gene signature for determining Ewing sarcoma prognosis [150], and in a study using Fourier transform infrared (FTIR) spectroscopy of post-treatment Ewing sarcoma tissue sections, machine-learning approaches predicted patient mortality in 92% of cases [151].

Though AI and machine learning have drawbacks in pathology, such as the limited amounts of tissues in biopsies, potential problems caused by focus [152], and the potential for user error in the acquisition of images and their interpretation. However, it is abundantly clear that use of AI can be very advantageous in analysis of complex data sets and identification of potential therapeutic targets. Whether they will become effective tools for diagnosis of pediatric sarcomas remains an unanswered question, as large studies have yet to be performed.

## 8. Conclusions

Conclusion: In spite of great strides made in the understanding and treatment of pediatric sarcomas, a significant subset remains recalcitrant to treatment. Several new research methods, including scRNAseq, spatial multi-omics, high throughput functional genomics, and CRISPR-Cas 9 technology, complimented by artificial intel-ligence and machine learning, offer the means to further delve into the biology of these cancers and to find possible new ways to attack them. A number of recent studies show the potential benefits of these new tools, but limitations in the availability of materials for analysis remains a setback that must be overcome.

## Figures and Tables

**Figure 1 cancers-14-02515-f001:**
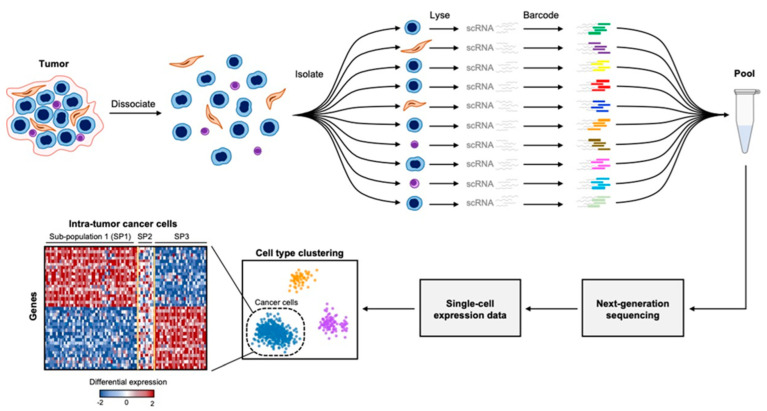
Overview of single-cell RNA sequencing. Tumor tissue is disaggregated into constituent single cells which can be isolated through various methods. Isolated single cells are lysed, and cellular RNA is converted into cDNA. cDNA from individual cells is ligated to cell-specific oligonucleotide barcodes after which all cDNA is pooled and sequenced collectively via next-generation sequencing. The transcriptomes of each cell are then segregated computationally using the barcode sequences, enabling the analysis of single-cell gene expression profiles.

**Figure 2 cancers-14-02515-f002:**
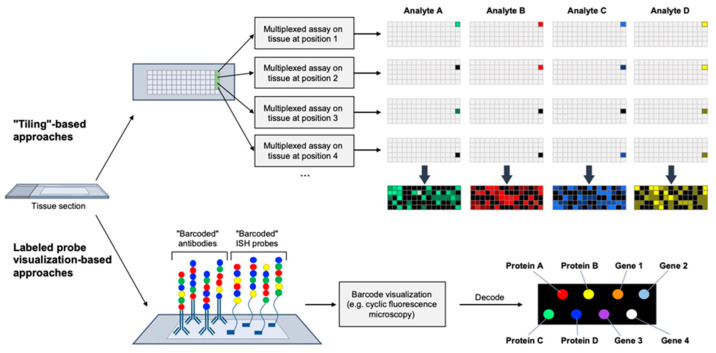
Overview of spatial genomics, transcriptomics, and proteomics methodologies. Tissue sections are typically visualized either through a “tiling”-based approach or through the visualization of “barcoded” probes.

**Figure 3 cancers-14-02515-f003:**
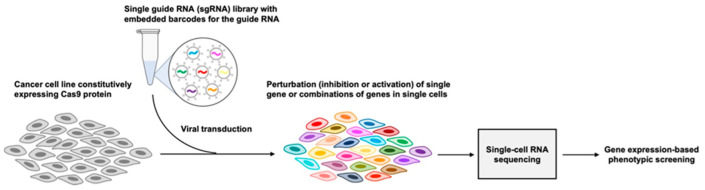
Schematic of high-throughput genetic screening using clustered regularly interspaced short palindromic repeats (CRISPR)-based perturbations and single-cell RNA sequencing. Tumor cell lines that constitutively express the Cas9 protein are infected with lentiviruses that each contain (1) a single guide RNA (sgRNA) specific for a single gene, and (2) a unique oligonucleotide barcode specific for the sgRNA. In the context of a CRISPR-i or CRISPR-a screen, the Cas9 protein expressed by the cell line has been modified to inhibit or activate transcription, respectively. In this context, the sgRNAs target the regulatory regions of genes. In the context of a gene knockout screen, however, the Cas9 protein expressed by the cell line cleaves the genomic region targeted by the sgRNA, rendering the targeted gene inactive. Following the CRISPR-based perturbation, cells can be further manipulated (e.g., treated with a drug) after which they undergo single-cell RNA sequencing. Using the sgRNA-specific and cell-specific barcodes, the expression profile associated with each perturbation can then be delineated.

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
