# Peer review of "Pediatric Sarcomas: The Next Generation of Molecular Studies"

_cancers, 2022, doi:10.3390/cancers14102515_

Round 1

Reviewer 1 Report

In the article entitled “Pediatric Sarcomas: the Coming Generation of Molecular Studies » by P. Giannikopoulos, the authors give a description of the new generation of molecular techniques being recently developed to study human neoplasia and discuss applications to the study of pediatric sarcomas.

Major comments:

  • The article is interesting and well-written and provides useful data concerning relatively new molecular approaches such as single-cell profiling, spatial omics, and high throughput functional genomics.
  • My main concern is that the paper does not focus enough on pediatric sarcomas and is rather too “general”. More examples of potential applications of these techniques to study main pediatric sarcoma subtypes should be explained and discussed.
  • In the introduction, a paragraph should be dedicated to the genetic features of pediatric sarcoma, with the description of the main molecular alterations (i.e chimeric transcription factors, mutations, epigenetic alterations, complex genomes) and the biological questions that remain to be addressed.
  • In the section dedicated to single cell approaches, recent references on major studies on pediatric sarcomas are lacking (Zhou et al, Nature Comm 2020; Aynaud et all Cell Reports 2020)
  • In the section High Throughout functional genomics, references and discussion on major studies focusing on pediatric sarcoma are lacking: Wang et al, Sci Adv 2021, Seong et al, Cancer cell 2021, Phelps et al PNAS 2016 and others!

Minor comments

  • Reference 24 in canine osteosarcoma is not appropriate. Reference to human osteosarcoma should be included and discussed (Zhou et al, Nature Comm 2020)

  • Some typos remain to be corrected including for example
    • Page 3 line 92 “PAX-FOXO1”
    • Page 3 line 107 “an sc-RNA-seq”

Author Response

  • My main concern is that the paper does not focus enough on pediatric sarcomas and is rather too “general”. More examples of potential applications of these techniques to study main pediatric sarcoma subtypes should be explained and discussed.
    • Reply: We have added a number of examples of uses of these techniques.
  • In the introduction, a paragraph should be dedicated to the genetic features of pediatric sarcoma, with the description of the main molecular alterations (i.e chimeric transcription factors, mutations, epigenetic alterations, complex genomes) and the biological questions that remain to be addressed.
    • Reply: A introductory section has been added with comments about the genetic features of childhood sarcomas.
  • In the section dedicated to single cell approaches, recent references on major studies on pediatric sarcomas are lacking (Zhou et al, Nature Comm 2020; Aynaud et all Cell Reports 2020).
    • Reply: These references have been added and discussed.
  • In the section High Throughout functional genomics, references and discussion on major studies focusing on pediatric sarcoma are lacking: Wang et al, Sci Adv 2021, Seong et al, Cancer cell 2021, Phelps et al PNAS 2016 and others!:
    • Reply: These reference have been added and discussed.
  • Reference 24 in canine osteosarcoma is not appropriate. Reference to human osteosarcoma should be included and discussed (Zhou et al, Nature Comm 2020)
    • This reference has been deleted and replaced with the suggested one.
  • Some typos remain to be corrected including for example
      • Page 3 line 92 “PAX-FOXO1”: CORRECTED
      • Page 3 line 107 “an sc-RNA-seq”: CORRECTED.

Reviewer 2 Report

This is an interesting, if short review of the authors perceived direction of molecular studies in paediatric sarcomas. Whilst it highlights some interesting studies and future directions it lacks enough background and synthesis of applications already occurring in the paediatric setting to make a compelling comprehensive review. In its current form it reads as more of a review of the technologies available rather than their application to paediatric sarcomas. The authors describe some of the more novel molecular techniques that are being employed in cancer research. It is well organised and clearly describes the topics of spatial multi-omics and high throughput functional genomics. However, it clearly lacks any synthesis of the use of these techniques in paediatric sarcomas and any discussion of the long-term clinical benefits of application of the techniques described for paediatric sarcomas. 

In order to create a more comprehensive and stand-alone review it would be helpful to include in the introduction more information/background about paediatric sarcomas such as the current treatments, long-term survival rates and risk stratification. They should also include some discussion of the low mutational burdens they mention later. 

The scRNAseq section lacks discussion of a number of papers that have recently emerged using this technology for paediatric cancers. Examples include Miller et al, 2020; Aynaud et al, 2020; Hong et al, 2021; Patel et al, 2021; Khoogar et al, 2022.

How to the authors envisage spatial multi-omic tools could and should be used for paediatric sarcomas? Limitations of these techniques should be discussed.

There are numerous examples of the use of CRISPR in paediatric sarcomas and these should be explored. 

Line 243: Change title from 5 to 6. AI/machine learning provides promising predictions for cancers, can authors talk about upcoming challenge and differences specifically in paediatric sarcoma compared to other types of cancers? This section should describe how the work that has already been done using the techniques described could be enhanced in future and should include more about how the authors think the technology they have described could be used for clinical benefit. 

Recommend ending with a dedicated conclusion paragraph. The authors should suggest how the new genetic technologies will benefit patients with paediatric sarcomas clinically.

Author Response

  • This is an interesting, if short review of the authors perceived direction of molecular studies in paediatric sarcomas. Whilst it highlights some interesting studies and future directions it lacks enough background and synthesis of applications already occurring in the paediatric setting to make a compelling comprehensive review. In its current form it reads as more of a review of the technologies available rather than their application to paediatric sarcomas. The authors describe some of the more novel molecular techniques that are being employed in cancer research. It is well organised and clearly describes the topics of spatial multi-omics and high throughput functional genomics. However, it clearly lacks any synthesis of the use of these techniques in paediatric sarcomas and any discussion of the long-term clinical benefits of application of the techniques described for paediatric sarcomas. 
    • Reply: We thank the reviewer for his comments, which assisted us in strengthening the manuscript.  We have added more examples of use of these techniques and additional discussion of their potential use and benefits in childhood sarcoma.
  • In order to create a more comprehensive and stand-alone review it would be helpful to include in the introduction more information/background about paediatric sarcomas such as the current treatments, long-term survival rates and risk stratification. They should also include some discussion of the low mutational burdens they mention later. 
    • Reply: We have added an introductory section that contains background on treatments, survival rates, risk stratification, and mutations.
  • The scRNAseq section lacks discussion of a number of papers that have recently emerged using this technology for paediatric cancers. Examples include Miller et al, 2020; Aynaud et al, 2020; Hong et al, 2021; Patel et al, 2021; Khoogar et al, 2022.
    • These references have been added and discussed.
  • How to the authors envisage spatial multi-omic tools could and should be used for paediatric sarcomas? Limitations of these techniques should be discussed.
    • Additional information and references have been added concerning use of multi-omics to explore tumor heterogeneity and microenvironment, and we have discussed some potential limitations to its use.
  • There are numerous examples of the use of CRISPR in paediatric sarcomas and these should be explored. 
    • The  CRISPR section has been re-worked, with additional references and discussion.
  • Line 243: Change title from 5 to 6. AI/machine learning provides promising predictions for cancers, can authors talk about upcoming challenge and differences specifically in paediatric sarcoma compared to other types of cancers? This section should describe how the work that has already been done using the techniques described could be enhanced in future and should include more about how the authors think the technology they have described could be used for clinical benefit. 
    • The titles have been re-numbered, and additional discussion and references have been added to the section on artificial intelligence.
  • Recommend ending with a dedicated conclusion paragraph. The authors should suggest how the new genetic technologies will benefit patients with paediatric sarcomas clinically.
    • A dedicated conclusion section has been added.

Reviewer 3 Report

This is a very interesting summary of modern technology potentially useful in research and later probably in routine diagnostics of the pediatric soft tissue sarcomas or more generally phrased, that of the pediatric solid tumours. It is a thought-provoking paper projecting the near future of oncology research, which seems to be wise to publish. Although I am a bit more cautious regarding the omnipotent nature of AI technology and its absolute benefit in Medicine, I consider the paper a useful reading material for everybody working in the field of oncology, diagnostics and research.

Author Response

This is a very interesting summary of modern technology potentially useful in research and later probably in routine diagnostics of the pediatric soft tissue sarcomas or more generally phrased, that of the pediatric solid tumours. It is a thought-provoking paper projecting the near future of oncology research, which seems to be wise to publish. Although I am a bit more cautious regarding the omnipotent nature of AI technology and its absolute benefit in Medicine, I consider the paper a useful reading material for everybody working in the field of oncology, diagnostics and research.

We thank the reviewer for his/her comments and hope that the additional text and references will further enhance its value.

Round 2

Reviewer 1 Report

All the major comments have been addressed.

Minor issues remain to be corrected in the introduction

Line 31:melanoma do not belong to epithelial tumors

Line 37-40: Wilms tumors and neuroblastoma are not considered as mesenchymal tumors/sarcomas and should not be cited here

Page 3 line 119-127:GIST is rather an adult disease and should not be described as an example of “pediatric sarcoma”.

Line 122:PDGFB should be corrected for PDGFRB

Line 127: why talking about immunotherapy? This is rather targeted therapy

Author Response

Line 31:melanoma do not belong to epithelial tumors

The term "epithelial tumor" has been deleted.

Line 37-40: Wilms tumors and neuroblastoma are not considered as mesenchymal tumors/sarcomas and should not be cited here

Wilms tumors and neurolblastoma have been deleted from this section.

Page 3 line 119-127:GIST is rather an adult disease and should not be described as an example of “pediatric sarcoma”.

The term "pediatric sarcoma" has been deleted.

Line 122:PDGFB should be corrected for PDGFRB

Corrected.

Line 127: why talking about immunotherapy? This is rather targeted therapy.

"Immunotherapy" has been deleted and replaced with "targeted therapy".
